

# Carbon leaks from flooded land: do we need to re-plumb the inland water active pipe?

Gwenaël Abril[1,2] and Alberto V. Borges[3]

[1] Biologie des Organismes et Ecosystèmes Aquatiques (BOREA), Muséum National d'Histoire Naturelle, 61 rue Buffon, 75231, Paris cedex 05, France.

[2] Programa de Geoquímica, Universidade Federal Fluminense, Outeiro São João Batista s/n, 24020015, Niterói, RJ, Brazil.

[3] Université de Liège, Unité d'Océanographie Chimique, Institut de Physique (B5a), B-4000, Belgium

MS for Biogeosciences Discussions, Article type: **Ideas and perspectives**



**ABSTRACT**
At the global scale, inland waters are a significant source of atmospheric carbon (C),
particularly in the tropics. The active pipe concept predicts that C emissions from
streams, lakes and rivers are largely fuelled by terrestrial ecosystems. The traditionally
recognized C transfer mechanisms from terrestrial to aquatic systems are surface runoff
and groundwater drainage. We present here a series of arguments that support the idea
that land flooding is an additional significant process that fuels inland waters with C at
the global scale. Whether the majority of $CO_2$ emitted by rivers comes from floodable
land (approximately 10% of the continents) or from well-drained land is a fundamental
question that impacts our capacity to predict how these C fluxes might change in the
future. Using classical concepts in ecology, we propose, as a necessary step forward, an
update of the active pipe concept that differentiates floodable land from drained land.
Contrarily to well-drained land, wetlands combine strong hydrological connectivity with
inland waters, high productivity assimilating $CO_2$ from the atmosphere, direct transfer of
litter and exudation products to water and waterlogged soils, a generally dominant
allocation of ecosystem respiration below the water surface and a slow gas exchange
rate at the water-air interface. These properties force plants to pump atmospheric C to
wetland waters and, when hydrology is favourable, to inland waters as organic C and
dissolved $CO_2$. This wetland $CO_2$ pump may contribute disproportionately to $CO_2$
emissions from inland waters, particularly in the tropics, and consequently at the global
scale. In future studies, more care must be taken in the way that vertical and horizontal
C fluxes are conceptualized along watersheds and 2D-models that adequately account
for the hydrological export of all C species are necessary. In wetland ecosystems,
significant effort should be dedicated to quantifying the components of primary
production and respiration in air, water and waterlogged soils, and these metabolic



rates should be used in coupled hydrological-biogeochemical models. The construction
of a global typology of wetlands also appears necessary to adequately integrate
continental C fluxes at the global scale.



## 1. INTRODUCTION

Continental surfaces play a major role on the present and past climates, in particular

through the exchange of greenhouse gases (GHGs) such as carbon dioxide ($CO_2$) and

methane ($CH_4$) with the atmosphere (Ciais et al. 2013). Conversely, the global climate

affects the continental carbon (C) budget, as biological productivity and the capacity of

ecosystems to store C are influenced by temperature, rainfall and other climatic

variables (Heimann and Reichstein 2008; Reichstein et al. 2013). The continental C

budget is in addition affected by direct human alterations such as

deforestation/reforestation and other land use changes. On continents, the C cycle is

tightly coupled to the water cycle, and $CO_2$ and $CH_4$ budgets strongly depend on how and

how much water circulates through the plants, soil, groundwater, and surface waters to

the coastal ocean. Biogeochemical processes and fluxes in the critical zone, the

permeable layer of the continents from the vegetation top to the aquifer bottom (Lin

2010), have varied drastically at geological time scales (Knoll and James 1987).

Emissions of GHGs from continental ecosystems are expected to be highly sensitive to

precipitation and hydrology in the future. Water is necessary for plant photosynthesis;

moisture strongly controls respiration in soils; the presence of water promotes

anaerobic conditions and $CH_4$ production in wetlands, while soil desiccation promotes

soil $CH_4$ oxidation. Water also considerably contributes to continental C budgets because

rivers transport C laterally; C being later trapped in sediments, emitted as $CO_2$ and $CH_4$

to the atmosphere, or exported to the ocean (Garrels and Macknezie 1971; Meybeck

1982; Cole et al. 2007).

In terms of $CO_2$ and $CH_4$ fluxes, continental landscapes act as a heterogeneous mosaic,

and some ecosystems store or emit more atmospheric C than others. Some small





surfaces can behave as hotspots and disproportionately contribute to the total C mass
balance at the regional, continental and global scales. Surface waters are recognized
hotspots for $CO_2$ and $CH_4$ fluxes (Cole et al. 1994; Cole and Caraco 2001; Bastviken et al.
2011; Raymond et al. 2013; Holgerson and Raymond 2016). Natural surface waters
include the open waters of streams, lakes, rivers and estuaries (approximately 3.5% of
the continents) as well as intermittently flooded land, where a canopy of vegetation is
active above the water and/or when water is temporarily absent: swamps, marshes and
floodplains, also called wetlands, that occupy approximately 10% of the continents
(Downing 2009). In general, inland waters and wetlands show higher atmospheric C
exchange rates per surface area than the surrounding land: Wetlands are recognized for
their high productivity, sedimentary organic carbon (OC) burial and $CH_4$ emissions
(Mitsch et al 2013). Inland waters (rivers, streams, lakes and reservoirs) act as a very
significant source of atmospheric $CO_2$ at the global scale (Raymond et al. 2013).

Although the magnitude of $CO_2$ outgassing from inland surface waters at the global scale
is still subject to large uncertainties, there is consensus that the quantity of C exported
from land to freshwaters (1.9-3.2 PgC yr$^{-1}$) was larger than the C flux ultimately reaching
the ocean (0.9 PgC yr$^{-1}$, Fig. 1b). Cole et al. (2007) have conceptualized inland waters as
an active pipe (Fig. 1b), receiving, processing, emitting, and storing terrestrial C during
its travel from land to the ocean, as opposed to a passive pipe that simply transports
terrestrial C conservatively to the ocean (Fig. 1a), as generally assumed in earlier
literature from the 1970's and 1980's (Garrels and Mackenzie 1971; Meybeck 1982).
Since this definition, it has been assumed that most of the C emitted by inland waters
was initially fixed upland by terrestrial vegetation, then transported from soils to
aquatic systems with runoff and drainage, and finally emitted downstream as $CO_2$ to the



atmosphere. Because no satisfactory methods are available yet to estimate directly the
flux of C across the land-water boundary (e.g., Deirmendjian et al. 2018), this flux is
calculated as the sum of outgassing from inland waters, burial in freshwater and
estuarine sediments, and export to the coastal ocean (Cole et al. 2007). However, the
processes controlling C fluxes at the land-water interface are poorly understood and
some potential inconsistencies could arise when comparing C budget derived from
terrestrial studies with those derived from aquatic studies. Here, we provide some
additional evidence demonstrating that the transfer of terrestrial C to rivers could occur
preferentially from floodable ecosystems. We suggest that wetlands behave not only as a
significant source of atmospheric $CH_4$ and a long-term C sink in soils (Mitsch et al. 2013)
but also as an efficient $CO_2$ pump that exports dissolved and particulate C to inland
waters. Using classical concepts in ecology, we analyse qualitatively and quantitatively
how ecosystem production and respiration affect C export from drained-land and from
wetlands. We stress that our current understanding of processes and our ability to
measure and quantify C metabolic and hydrological fluxes must be considerably
improved to understand the origin of carbon in inland waters and predict future
continental GHG budgets in the mosaic of continental ecosystems.

**2. CONCEPTUALIZING AND FORMULATING C FLUXES**
Fluxes of C through the boundaries of an ecosystem, *i.e.,* vertical exchange with the
atmosphere and burial in soils and sediments on the one hand, and horizontal exchange
between lands, wetlands and aquatic ecosystems on the other hand, are driven by
metabolic processes in each ecosystem and physical processes that transport C such as
hydrology, wind, turbulent mixing, sediment deposition/resuspension, etc. Following





the conventions of Chapin et al. (2006), the net $CO_2$ exchange of an ecosystem with the
atmosphere is partitioned into several forms of C fluxes (Fig. 2):

$-NEE = NECB + F_{other} + E$ (Eq. 1)

where NEE is net ecosystem exchange (the net $CO_2$ flux from the ecosystem to the
atmosphere), NECB is the net ecosystem carbon balance (the net C accumulation in the
ecosystem), $F_{other}$ is the sum of vertical fluxes of volatile forms of C other than $CO_2$ ($CH_4$,
carbon monoxide, volatile organic carbon) from the ecosystem to the atmosphere and E
is horizontal C export by hydrological transport, trading of food, feed and wood (Ciais et
al. 2008). Among the components of E, only hydrological horizontal transport of C will
be discussed in this paper. Note that, by convention, NEE is opposite in sign to NECB
because NEE is defined by atmospheric scientists as a C input to the atmosphere,
whereas NECB is defined by ecologists as a C input to ecosystems (Chapin et al. 2006).

Regarding metabolic fluxes, net ecosystem production (NEP) is defined as:

$NEP = GPP-ER$ (Eq. 2)

where GPP is gross primary production and ER is ecosystem respiration. For conceptual
and methodological reasons, it is necessary to consider separately the autotrophic and
heterotrophic components of ER as:

$NEP = GPP-AR-HR$ (Eq. 3),

$NPP = GPP – AR$ (Eq. 4), and:

$NEP = NPP – HR$ (Eq. 5)

where AR and HR are, respectively, the autotrophic and the heterotrophic components
of ER and NPP is net primary production. A positive NEP (Eq. 2) reduces the
concentration of $CO_2$ and/or dissolved inorganic carbon (DIC) inside the ecosystem and
generates a gradient that causes atmospheric $CO_2$ to enter the ecosystem. One process





that makes -NEE diverge from NEP and NECB is when significant amounts of inorganic C
enter or leave the ecosystem as DIC in the aquatic phase with horizontal hydrological
transport rather than through atmospheric exchange (Chapin et al. 2006). In addition to
this divergence between -NEE and NEP, NECB deviates from NEP when C enters or
leaves the ecosystem in forms others than $CO_2$ or DIC (Eq. 1). This includes horizontal
transport of particulate and dissolved OC by hydrological processes, as well as vertical
$CH_4$ fluxes, a secondary C flux that is significant for the active pipe concept, as well as for
climate regulation.

An adequate conceptualization of atmospheric C fluxes along watersheds implies first
the definition of functional boundaries inside the boundless C cycle (Battin et al. 2009),
at least between three types of ecosystems that have fundamentally different properties
with respect to atmospheric $CO_2$ (Fig. 2): (1) the terrestrial, never flooded land and its
biosphere (forest, crops, shrub, grassland and their well-drained soils and
groundwater); (2) the floodable land and its mosaics of wetlands with extremely
variable ecological and hydrological properties; (3) the open waters of streams, lakes
and rivers. Some estimations of $CO_2$ outgassing from inland waters have included
wetland surface areas (Richey et al. 2002; Aufdenkampe et al. 2011; Sawakuchi et al.
2017), while some others have not (Cole et al. 2007; Tranvik et al. 2009; Raymond et al.
2013). However, wetlands are functionally different from inland waters because their
canopy of vegetation can alter the direction of atmospheric $CO_2$ exchange (Raymond et
al. 2013; Abril et al. 2014). Assuming that the $CO_2$ flux at the water-air interface
equals -NEE in wetlands (Richey et al. 2002) implicitly supposes that GPP and the aerial
compartment of AR (Fig. 2b) are null or exactly balanced, which is incorrect. A functional
definition of wetlands with respect to C cycling could be the *intermittent and/or*



*vegetated flooded land*, in contrast with the well-drained land which is never flooded and
whose topsoil is never waterlogged, and with the permanent and open waters of lakes
and rivers without emerged or floating vegetation. This definition allows clear
delineation of the three sub-systems using remote sensing (e.g., Melack and Hess 2010)
and is also functional with respect to the conceptualization and quantification of C
cycling (Fig. 2).

Second, our conceptual model should be two-dimensional (vertical and up-downriver),
and should consider the hydrological net export term E in Eq. 1 as a potentially
significant component of -NEE and NECB (Fig. 2), in accordance with the active pipe
concept. In well-drained terrestrial ecosystems, surface runoff and drainage export C to
inland water, and E is necessarily always positive. In inland water and wetlands, E must
be conceptualized and quantified as the net balance between hydrological import to and
export from the ecosystems and, depending on each case, E can be positive or negative.
In fact, C fluxes along watersheds must be seen as a cascade from one sub-system
upstream to another sub-system downstream. Several chemical forms of C are involved
in the E term, which can be written as the sum of the export of four terms:
$\quad\quad\quad\quad E = E_{POC}+E_{DOC}+E_{CO2}+E_{CH4}$ (Eq. 6)
Particulate and dissolved organic C (POC and DOC) are derived from NPP; DIC is in part
the result of ER, that release dissolved $CO_2$ (as well as $CH_4$) to waters and in part the
result of chemical weathering that generates alkalinity. Weathering of carbonate and
silicate rocks is mediated by soil $CO_2$ derived from respiration, so that weathering is also
a component of ER. Because chemical weathering is assumed to occur mostly upland,
alkalinity is considered as a relatively conservative chemical form of river C, although
some exceptions have been reported in floodplains of tropical rivers (Boucher et al.





2012; Geeraert et al. 2017). Here, we will discuss only the fraction of DIC that occurs as
excess $CO_2$, that is, the DIC that is potentially lost after complete water-air equilibration
(Abril et al. 2000). Concerning dissolved $CH_4$, the role of wetlands was identified in the
literature for sustaining $CH_4$ emissions in adjacent rivers (Borges et al. 2015b) and lakes
(Juutinen et al. 2003). However, owing to its low solubility and the fact that emissions
from wetlands occur mostly as ebullition or through plants (contributing to the $F_{other}$
term in Fig. 2B), the contribution of $E_{CH4}$ to E is small (few percent) in most ecosystems.

NEE is generally negative in forests (Luyssaert et al. 2010; Ciais et al. 2013) and
wetlands (Morison et al. 2000; Saunders et al. 2007; Lu et al. 2016) but positive in lakes
and rivers (Cole et al. 1994; 2007; Raymond et al. 2013) (Fig. 3). Compared to NEE,
exchange of $CH_4$ with the atmosphere ($F_{other}$ in Eq. 1) is significant in wetlands but not in
forests (Ciais et al. 2013; Saunois et al. 2016) and probably not in inland waters. Indeed,
budgets of $CH_4$ emissions from inland waters strongly depend on whether wetland areas
were included or not and, in general, open waters of rivers and lakes emit $CH_4$ at rates
approximately 100 times lower than $CO_2$ (Melack et al. 2004; Bastviken et al 2011;
Borges et al. 2015a). The occurrence of a horizontal transport of C by streams and rivers
implies a positive E term in terrestrial ecosystems, where -NEE should exceed NECB.
The E is probably also large in wetlands, where -NEE likely exceeds net storage in soils
plus $CH_4$ emissions (Eq. 1; Fig. 1c). In contrast, in aquatic systems NECB exceeds -NEE
and E is negative (Cole and Caraco 2001; Battin et al. 2008) because these ecosystems
receive in general more C from upstream than they export downstream. In addition, the
fact that part of E occurs as OC implies that NEP exceeds NECB in terrestrial systems and
wetlands that export OC, whereas NECB will exceed NEP for instance in lakes or
estuaries that receive and store large amounts of allochthonous OC in their sediments





(Lovett et al. 2006; Cole et al. 2007; Tranvik et al. 2009). In general, C fluxes at the
boundaries of ecosystems and metabolic fluxes inside the ecosystems suggest that the
magnitude of the export term E in Eq. 1 and Fig. 2 and the deviation of -NEE from NECB
and from NEP, will strongly depend on their hydrological connectivity, together with the
allocation of GPP and ER in air and water.




**3. THE INLAND WATER PERSPECTIVE**
Global estimates of $CO_2$ emissions from inland waters (Cole et al. 1994; Raymond et al.
2013; Lauerwald et al. 2015) are derived from $CO_2$ flux intensities computed from the
water-air gradient of the partial pressure of $CO_2$ ($pCO_2$) and the gas transfer velocity at
the water-air interface and scaled to the surface area of lakes and rivers. Each of the
three terms suffers for uncertainties and generally poor data coverage. Cole et al. (1994)
provided the first quantification of the $CO_2$ emission to the atmosphere from lakes (0.1
PgC yr$^{-1}$), which was later confirmed by an updated calculation by Sobek et al. (2005).
Cole and Caraco (2001) estimated global $CO_2$ degassing for rivers and streams, which
has been recently re-evaluated by Raymond et al. (2013) and Lauerwald et al. (2015).
The two latter studies are based on $pCO_2$ computed from pH and alkalinity from the
same database (GLORICH, Hartmann et al. 2014) but with different data selection
criteria and scaling approaches. Raymond et al. (2013) extrapolated discrete $pCO_2$
values per COSCATS catchment aggregated units (Meybeck et al. 2006) and obtained a
global $CO_2$ emission to the atmosphere of 0.3 PgC yr$^{-1}$ from lakes and 1.8 PgC yr$^{-1}$ from
rivers and streams. A potential problem in this estimation comes from the calculation of
$pCO_2$ from pH and alkalinity, which greatly overestimates $pCO_2$ (up to several hundred
percent) in many acidic organic rich "black" waters such as those found in the tropics
and the boreal zone (Abril et al. 2015). Lauerwald et al. (2015) computed river $pCO_2$
values on a regular grid (1°x1°), using a multiple regression model based on the
GLORICH $pCO_2$ data and modelled terrestrial NPP on the catchment, population density,
air temperature and slope; this method provided a lower estimate of global $CO_2$
emission for rivers of 0.7 PgC yr$^{-1}$. The strong divergence of global $CO_2$ emission
estimates in these two studies most likely reflects the low data coverage in tropics that
account for nearly 80% of the modelled global emission, although in the GLORICH



database nearly all of the data in the tropics are from the Amazon. Recent direct $pCO_2$
measurements in several African rivers (Borges et al. 2015a), and in the Amazon (Abril
et al. 2014) scaled to the tropics with wetland coverage (Borges et al. 2015b) provide a
value of $1.8\pm0.4$ PgC yr$^{-1}$ of $CO_2$ outgassing from tropical rivers alone (latitude < 25°),
and thus support the higher estimate of Raymond et al. (2013). A larger estimate of the
global river $CO_2$ outgassing of 3.9 PgC yr$^{-1}$ has been published recently (Sawakuchi et al.
2017). However, we choose not to consider this number in our analysis because it is
based on observations in the Amazon River that include the floodplain areas with a
canopy of vegetation above the water.

According to the active pipe concept (Fig. 1b), the emission of $CO_2$ to the atmosphere
from inland waters is attributed to terrestrial C fixed by plants on the catchment. The
transfer occurs as (1) an input of dissolved $CO_2$ (and $CH_4$) originating from soil
respiration, that will be further degassed from waters ($E_{CO2}$ and $E_{CH4}$ in Eq. 6); (2) an
input of particulate and dissolved organic C ($E_{DOC}$ and $E_{POC}$) followed by heterotrophic
degradation to $CO_2$ and $CH_4$ in the aquatic system (Del Giorgio et al. 1999; Prairie et al.
2002; Cole et al. 2000; Battin et al. 2008; Hotchkiss et al. 2015). Inland waters,
particularly lakes, also store significant quantities of OC mainly of terrestrial origin in
their sediments (Cole et al. 2007; Tranvik et al. 2009). In aquatic systems, all the GPP
and ER occur in water and sediments (Fig. 2c) and can be quantified with *in vitro* or *in*
*situ* incubations. In addition, the $CO_2$ outgassing flux measured with floating chambers in
open waters give a direct estimate of -NEE (although this method may create artefacts at
the water-air interface), and diurnal changes in water $pCO_2$ (or oxygen concentration)
can provide an estimate of GPP and ER. Battin et al. (2008) made a global synthesis of
aquatic metabolism rate measurements and confirmed that stream, river and estuarine





ecosystems are overall net heterotrophic and respire a total flux of 0.3 PgC $yr^{-1}$. The fact
that net heterotrophy (negative NEP) is in general lower than $CO_2$ outgassing in inland
waters, led Hotchkiss et al. (2015) to differentiate "internal $CO_2$" (from –NEP) from
"external $CO_2$" coming from groundwater inputs of DIC. Indeed, inputs of groundwater
DIC are acknowledged as sustaining a significant fraction of the $CO_2$ emissions from
lakes (Butman and Raymond 2011; McDonald et al. 2013) and from rivers, especially
headwaters (Johnson et al. 2008; Hotchkiss et al. 2015; Deirmendjian and Abril 2018).
Horizontal transfer of respiration-derived DIC from terrestrial and wetland ecosystems
to aquatic ecosystems explain why aquatic NEE ($CO_2$ outgassing) greatly exceeds –NEP
(negative NEP, net heterotrophic ecosystems) in rivers (Abril et al. 2014; Hotchkiss et al.
2015; Borges et al. 2015a). Conversely, this outgassing flux from aquatic systems implies
that in terrestrial ecosystems and wetlands that release DIC laterally, NEP exceeds -NEE.




## 4. THE TERRESTRIAL PERSPECTIVE

Hydrological C export as a significant loss term for terrestrial ecosystems has been

considered in more detail only relatively recently (e.g., Ciais et al. 2008) and is included

in only a very limited number of global terrestrial models (Krinner et al. 2005).

Terrestrial C budgets at the plot and the continental scales are based on different

methods not consistent and precise enough to estimate hydrological C export as a

residual flux. In addition, no direct standardized experimental method is available yet to

directly estimate the flux of C across the boundary between land and water, and the E

term in Eq. 1 for terrestrial systems is almost always calculated from a C mass balance in

inland waters (Fig. 1b; Ciais et al. 2013). Terrestrial -NEE calculated as the difference

between land use change and net land C flux is estimated at 2.6 PgC yr$^{-1}$ for the years

2000s (Ciais et al. 2013). In a conceptual model that ignores the different functionalities

between floodable and drained land (Fig. 1b), depending on what estimates are used for

the outgassing term (Raymond et al. 2013; Lauerwald et al. 2015) and for the sediment

burial term (Cole et al. 2007; Tranvik et al. 2009), the hydrological export necessary to

balance the inland water C budget is 1.9-3.2 PgC yr$^{-1}$, which corresponds to 75-125% of

the present net atmosphere-land C flux (Fig. 1b). The atmosphere-land net C flux of 2.6

PgC yr$^{-1}$ is derived from multiple approaches including atmospheric $CO_2$ inversion,

terrestrial ecosystem models and forest inventories (Ciais et al. 2013). The atmospheric

$CO_2$ inversion method integrates large continental areas that include inland waters.

Thus, the global -NEE calculated from continental-scale inversion models accounts for

$CO_2$ outgassing from inland waters. Intriguingly, the results of inversion methods are

relatively consistent with forest inventories and process-based models that do not

necessarily account for hydrological export (Ciais et al. 2013). However, when a





comparison is made at the plot scale with eddy-covariance data, model performance is
generally poor (Schwalm et al. 2010), and for instance modelled GPP can be
overestimated by more than 100% in tropical forests (Stöckli et al., 2008). If a -NEE
from atmospheric inversion is assumed close to NECB from inventories and process-
based models, then the E term (Eq. 1) is expected to be small, within the error of flux
estimates from the terrestrial perspective. If outgassing of $CO_2$ from freshwater is
already included in -NEE calculated by atmospheric inversion methods, and if this -NEE
value (2.0-3.0 PgC $yr^{-1}$) is very close to that of NECB (1.8-2.3 PgC $yr^{-1}$), then terrestrial
ecosystems cannot export the 0.6-1.0 PgC $yr^{-1}$ of recalcitrant OC that is buried in inland
waters (0.2-0.6 PgC $yr^{-1}$) and exported to the ocean (0.4 PgC $yr^{-1}$).

Spatially, global forest carbon accumulation occurs in boreal and temperate regions,
whereas tropical forests were found to be near neutral, with net emissions from land
use change being compensated by sinks in preserved tropical forests (Pan et al. 2011).
In contrast, Lauerwald et al. (2015) estimated that 78% of global $CO_2$ outgassing by
rivers occurred at a latitude lower than 25°. Such latitudinal uncoupling between $CO_2$
uptake by forests and $CO_2$ outgassing from rivers and lakes is intriguing and merits an
explanation. Indeed, it would imply that different climatic and/or anthropogenic forces
are driving these continental fluxes, in contradiction with the positive spatial correlation
between river $pCO_2$, air temperature and terrestrial NPP at the global scale (Lauerwald
et al. 2015). It should not be forgotten, however, that these correlations could be
indirect. Indeed, field $pCO_2$ data in the Amazon and in African rivers including the Congo,
reveal a strong positive influence of flooding and the presence of wetlands on water
$pCO_2$ (Abril et al. 2014; Borges et al. 2015a,b).



In terrestrial systems, few local studies at the plot scale compare -NEE or NECB
measurements with E derived from groundwater, spring and/or stream sampling. These
studies lead to very different conclusions from those of global modelling studies. In
remnant mature forests of Para, Brazil, Davidson et al. (2010) estimated the export of
dissolved $CO_2$ from soil and groundwater to streams at a value of 2-3 orders of
magnitude lower than the soil respiration and NPP. In temperate climate, Kindler et al.
(2011) quantified C leaching by combining a soil-water model and dissolved C analysis
in soil water; these authors reported significant E flux in croplands (25% of NECB),
grasslands (22%) but not in forests (less than 3%). In a temperate, forested and well-
drained watershed, Deirmendjian et al. (2018) monitored dissolved C concentrations in
groundwater and streams and estimated a total export E of 2% of -NEE as measured by
eddy-covariance at the same site. These modest export rates from forests in this limited
number of studies appear contradictory with the necessity of a large E term from
terrestrial ecosystems (1.9-3.2 PgC $yr^{-1}$ in Fig. 1b) to fuel inland waters at the global
scale (Cole et al. 2007; Ciais et al. 2013).

From an ecological point of view, a modest hydrological C export from well-drained
lands is also supported by the nature of their NEP components and more specifically by
the allocation of GPP and ER between air and water (Fig. 2,3). In terrestrial systems, GPP
assimilates atmospheric $CO_2$, and AR releases $CO_2$ partly in air (ARa), as foliar
respiration, woody tissue respiration, and partly in soil (ARs), as root respiration. HR
occurs almost entirely in soils (HRs). In forests, belowground respiration generally
accounts for 30-80% of ER, and aboveground respiration accounts for the remaining
fraction of ER (Davidson et al. 2006). Belowground respiration in soils (ARs and HR)
produces $CO_2$ mainly in superficial well-drained soils, where root density is highest and





which are enriched in biodegradable organic matter by litter fall and root exudation
(Ryan and Law 2005). When the land is well-drained, this $CO_2$ is released in the
unsaturated zone of the soil and mostly returns to the atmosphere across the soil-air
interface. In a tallgrass prairie, downward transfer of soil $CO_2$ to groundwater was only
approximately 1% of the soil-air $CO_2$ efflux (Tsypin and Macpherson 2012). For this
reason, $CO_2$ efflux from soils as measured with static chambers (Fig. 3) is commonly
used as an integrative measure of soil respiration (Ryan and Law 2005; Davidson et al.
2006) and until now, by considering the loss of $CO_2$ that dissolves in groundwater as
negligible or within the error of estimation of metabolic flux at the ecosystem scale.

The transfer of C from well-drained terrestrial ecosystems to aquatic systems (Fig. 3)
occurs through mechanical erosion of superficial soil by runoff that mobilizes POC
including young litter, more refractory mineral-bound OC, as well as dissolved humic
OC, and percolation of rainwater through soils that dissolves gaseous $CO_2$ and soil OC
and liberates DIC and DOC in groundwater, which is further drained to streams and
rivers. The fraction of HR that occurs in groundwater is probably modest in well-drained
ecosystems, as the deepest water-saturated soil horizons contain much less
biodegradable organic matter than the superficial soil (Ryan and Law 2005;
Deirmendjian et al. 2018). A modest export rate from forests is thus consistent with the
allocation of forest metabolism (in particular ER) mainly above the water table (Fig. 2a),
and with only few percent of -NEE ultimately reaching the aquatic system in non-
flooding conditions (Fig. 3).




**5. THE WETLAND PERSPECTIVE**

Even though wetlands cover an area of only approximately 10% of land surface

(Downing 2009), they act as hotspots of productivity and $CH_4$ emissions (Saunois et al.

2016). In addition, wetlands have strong hydrological connections with streams, rivers

and lakes. Ecologists formulated the hypothesis of wetlands as efficient C-exporters long

ago. Mulholland and Kuenzler (1979) reported several-fold higher DOC export from

swamps than from the surrounding landscape in North Carolina (US). Junk (1985)

described floodplain wetlands as a source of POC for the Amazon River; Wetzel (1992)

named littoral wetlands of lakes has "metabolic gates" for nutrients and organic C

between terrestrial and aquatic ecosystems. More recently, using a landscape ecological

approach, Jenerette and Lal (2005) commented on the determinant influence of

hydrology on wetland C fluxes, including downstream export to open waters.

Consequently, hydrological variation (the second dimension of the conceptual 2D-

Model) was identified as a factor of large uncertainty in wetland C cycling (Jenerette and

Lal 2005). Indeed, current available quantitative information on the C export flux (Eq. 6)

is particularly scarce. In wetlands, the quantification of metabolic C fluxes, and the

understanding of biogeochemical processes regulating -NEE, NEP, ER, and NECB have a

high degree of uncertainty. The partitioning of wetland community metabolism between

air, water and sediment, and the complex biological and physical processes that transfer

C in gaseous, dissolved, and particulate forms between these three sub-compartments

are only partially understood (e.g., Hamilton et al. 1995); they are also highly variable in

time and space, and difficult to measure in practice.

The few estimates of wetland C fluxes at the global scale strongly vary depending first on

the surface area considered for upscaling (Fig. 1c). Lenher and Döll (2004) calculated a



wetland surface area of 9-11 $10^6$ km², Mitsch et al. (2013) have used a value of 7 $10^6$
km², and Downing (2009) re-evaluated the total wetland area including smaller systems
to 13-16 $10^6$ km². Based on remote sensing data, Papa et al. (2010) provide a mean total
surface area of 3.4 $10^6$ km², with 56% located in the tropics, in agreement with previous
estimates by Pringent et al. (2001; 2007). More recently, Lu et al. (2016) use a larger but
probably unrealistic value of 33 $10^6$ km². Global wetland C fluxes consist in three major
terms in Eq. 1: (1) -NEE obtained from eddy-covariance measurements was up-scaled to
a value of 3.2 PgC yr$^{-1}$ (Lu et al. 2016), an estimate that needs to be corrected to 1.3 PgC
yr$^{-1}$ when applying the surface area re-evaluated by Downing (2009); in addition, the
arithmetic mean of available eddy covariance data (Lu et al. 2016) is probably not the
most appropriate way to upscale -NEE at the global scale, and a more precise typology of
wetland -NEE is necessary, based for instance on the classification of Lehner and Döll
(2004). (2) NECB is assumed as equal to organic C sequestration in soils and estimated
from $^{210}$Pb and $^{137}$Cs core dating (Mitsch et al. 2013), a method that ignores slow decay
in the soil C pool and can result in unrealistically high soil C sequestration rates
(Bridgham et al 2014); Indeed, Mitsch et al. (2013) proposed a global C sequestration
value of 0.8 PgC yr$^{-1}$, whereas Bridgham et al. (2014) re-evaluated this value to less than
0.1 PgC yr$^{-1}$. (3) The $F_{other}$ term for wetlands is mainly composed of $CH_4$ emissions and
estimated from bottom-up approaches using static chambers and process-based models
(Mitsch et al. 2013; Saunois et al. 2016), and top-down inversion models based on
atmospheric data (Saunois et al. 2016). Recent published estimates for the global
wetland $CH_4$ flux range between 0.2 PgC yr$^{-1}$ (Saunois et al. 2016) and 0.6 PgC yr$^{-1}$
(Mitsch et al. 2013). Wetland C sources and sinks are thus subject to large uncertainties
but still support the possibility of a residual C flux able to contribute significantly to
river C budgets at the global scale (Fig. 1c.).




Eddy covariance reveals strong negative NEE ($CO_2$ sink) in most wetlands (Morison et al.
2000; Jones and Humphries 2002; Saunders et al. 2007; Lu et al. 2016). However, if
wetland E is ignored but significant, GPP, NPP, NEP, and NECB deduced from the diurnal
changes of eddy $CO_2$ fluxes (Lu et al. 2016) would be overestimated and, inversely, ER
would be underestimated (Eqs.1-6). This point is particularly crucial because in
wetlands the aerial compartment contains most of the photosynthetic parts of the
ecosystem (GPP, NPP) fixing $CO_2$ directly from the atmosphere, whereas the aquatic
compartment contains the respiratory parts of the ecosystem (ER, HR and a large
fraction of AR) releasing $CO_2$ to waters but only part of it back to the atmosphere
because of gas-exchange limitation at the water-air interface (Fig. 3). Wetland 1D mass-
balance budgets also include an estimation of NPP, based on biomass inventories
(Mitsch et al. 2013; Sjögersten et al. 2014). One problem with NPP data is not accounting
for all the C transferred by the plant from the atmosphere to soil and waters. As the sum
of NEP and HR (Eq. 5), NPP does not include the fraction of GPP that is recycled by AR,
and most importantly, the root respiration in sediment and water, which is highly
significant below floating plant meadows (Bedford et al. 1991; Hamilton et al. 1995) and
in flooded forest (Piedade et al. 2010). Total AR should be divided into three
components according to:

AR=ARa+ARw+ARs        (Eq. 7)

where ARa, ARw and ARs are the fraction of AR occurring in air, water and soils,
respectively (Fig. 3). In wetlands, a canopy of vegetation protects the water-air interface
from wind stress and the gas transfer velocity is lower compared to surrounding open
waters (Foster-Martinez and Variano 2016; Ho et al. 2018). Consequently, only a limited
fraction of ARw and ARs will contribute to the $CO_2$ fluxes measured with static chambers



in wetlands. This is a second reason why wetland mass balances are incomplete and
may artificially shift wetlands to atmospheric C sources or sinks (Sjögersten et al. 2014).

The allocation of C stocks and metabolism above and below water is fundamentally
different in wetlands compared to well-drained land, and this considerably modifies
their ecological functionalities (Fig. 2 and 3). Although some wetland plants also use DIC
from water for photosynthesis, a large majority of wetland GPP is made by the emerged
part of plants that fix atmospheric $CO_2$ during the emersion periods, and/or during the
flooding because of their emerged or floating canopies (Piedade et al. 1994; Parolin et al.
2001; Engle et al. 2008). A large fraction (excluding wood) of the wetland biomass
produced annually is transferred directly to water and sediment as litter fall and fine
root production, where it fuels HR, including methanogenesis. Albeit important for $CH_4$
oxidation (Segarra et al. 2015), this leads to a $F_{other}$ (Eq. 1) as $CH_4$ fluxes more
significantly in wetlands than in well-drained terrestrial ecosystems (Ciais et al. 2013;
Saunois et al. 2016). In addition, because of anaerobic conditions in their soils, water-
tolerant plants can develop morphological aeration strategies (Haase and Rätsch 2010)
that actively transport oxygen to the root zone and enhance respiration and the release
of dissolved $CO_2$, $CH_4$ and other fermentative organic compounds such as ethanol to
waters and pore waters (Bedford et al. 1991; Hamilton et al. 1995; Piedade et al. 2010).
Plants also transport $CH_4$ directly from sediments to the atmosphere (Byrnes et al.
1995). This is why wetland water below plant canopies is generally hypoxic and highly
supersaturated in $CO_2$ (Bedford et al. 1991; Abril et al. 2014) and $CH_4$ (Hamilton et al.
1995; Borges et al. 2015b). Because the water-air interface behaves as a strong physical
barrier for gas diffusion, depending on hydrological features, dissolved $CO_2$ from



swamps, marshes and floodplains waters can be transported downriver for long
distance before being emitted to the atmosphere (Abril et al. 2014; Borges et al. 2015b).

The process that captures atmospheric $CO_2$ and exports organic and inorganic C to
rivers and lakes can be called the *wetland $CO_2$ pump*. This biological pump is also
consistent with chamber measurements that generally identify $CO_2$ sinks in vegetated
flooded areas and $CO_2$ sources in adjacent open waters (Pierobon et al. 2011; Ribaudo et
al. 2012; Peixoto et al. 2016). It is worth noting that little is known on how wetland -NEE
is affected by hydrology. For instance, a swamp of papyrus (*Cyperus papyrus*) on a
sheltered shore of Lake Naivasha, Kenya, was a $CO_2$ sink during immersion but a $CO_2$
source during emersion, when large amounts of plant detritus accumulated in soils were
exposed to air (Jones and Humphries 2002). In contrast, in the more hydrologically
dynamic Amazon floodplain, Brazil, a stand of *Echinochloa polystachya*, another C4 plant,
was a $CO_2$ sink during both immersion and emersion (Morison et al. 2000). This suggests
that a more efficient hydrological export of C in Amazon floodplains compared to Lake
Naivasha could have promoted an annual negative NEE (Eq. 1). Such competition
between C export and burial is also consistent with the more efficient C sequestration in
low flow-through wetlands (Mitsch et al. 2014).

Concerning wetland metabolic C balance, the fraction of OC produced by NPP that is not
respired *in situ* or buried in the wetland soil is exported to rivers systems as OC (Fig. 3),
according to:

$NPP = B + HR + E_{POC} + E_{DOC}$ (Eq. 8)

$NEP = B + E_{POC} + E_{DOC}$ (Eq. 9)



where B is the OC burial in the wetland soil. Thus, the export of POC and DOC from
wetlands is expressed as:

$E_{POC}+E_{DOC}$ = NEP – B = NPP – HR – B (Eq. 10)

Downstream, this organic material will undergo intense degradation in inland water
(negative NEP), contributing to $CO_2$ outgassing through the OC detrital pathway (Cole
and Caraco 2001; Battin et al. 2008).
Plants and microbes respiring in water, sediments, and the root zone (ARw and ARs and
HR) release dissolved $CO_2$ in wetland water. ARa is the only component of ER not
contributing to $E_{CO2}$. The fraction $\alpha$ of wetland ER occurring in water and sediment
(ARw and ARs) and almost all of the microbial HR (Eq. 11), release dissolved $CO_2$ (and
$CH_4$) to waters:

$\alpha$ER= ARw+ARs+HR        with (0<$\alpha$<1)        (Eq. 11)

part of these dissolved gases are emitted to the atmosphere, and another part is
exported by the water flow:

$\alpha$ER= $FCO_2$+$FCH_4$+$E_{CO2}$+$E_{CH4}$ (Eq. 12)

with            $E_{CO2} = \alpha\beta$ER and $F_{CO2} =\alpha(1 - \beta)$ER and (0<$\beta$<1) (Eq. 13)
For simplification, we do not include $E_{CH4}$ in this last equation because this term is
assumed to be modest (few %) compared to $E_{CO2}$. Indeed, the $\beta$ term might be much
smaller for $CH_4$ than for $CO_2$ due to preferential $CH_4$ ebullition and transport through
plants in wetlands (Chanton and Whiting 1995). For $CO_2$, the fraction $\beta$ depends on
hydrological and geomorphological parameters such as water depth, velocity and gas
exchange in the wetland. Using a simple model of lateral dissolved gas transport (Abril
et al. 2014), typical values of 1 cm $s^{-1}$ for the gas transfer velocity (Foster-Martinez and
Variano 2016; Ho et al. 2018) and 5000 ppmv for water $pCO_2$, we calculated a $\beta$ value of





0.93 for a water column of 1 m-depth flowing at a velocity of 10 cm s$^{-1}$ in a 100 m-long
wetland. When the water depth is established at 0.1 m instead of 1 m or the water
velocity is established at 1 cm s$^{-1}$ instead of 10 cm s$^{-1}$, $\beta$ decreases to 0.53. Consequently,
a large majority of the $CO_2$ produced by wetland below-water respiration is outgassed to
the atmosphere outside of the wetland. Finally, accounting for all terms in Eq. 6 in
wetlands leads to total export expressed as:
$E = (E_{DOC}+E_{POC})+(E_{CO2}+E_{CH4})=(NPP–HR–B)+(\beta\alpha ER -FCO_2-FCH_4)$          (Eq. 14)
$E = (E_{DOC}+E_{POC})+(E_{CO2}+E_{CH4})=(NPP–HR–B)+(\beta(ARw+ARs+HR)-FCO_2-FCH_4)$   (Eq. 14)
$E = NPP-B+\beta ARw+\beta ARs+(\beta − 1)HR-FCO_2-FCH_4$          (Eq. 15)
The correct 2D wetland mass balance budget in wetlands is also calculated as:
$NPP+\beta ARw+\beta ARs-(1 − \beta)HR =B+F_{CO2}+F_{CH4}+E$          (Eq.16).
The three terms ARw and ARs and HR together with the E term, are generally neglected
in wetland C budgets (Mitsch et al. 2013; Sjögersten et al. 2014).

WHAT TOOLS DO PLUMBERS NEED?
Quantifying hydrological C export from wetlands at the community, ecosystem, regional,
and global scales would require information that to date is still missing or incomplete.
General recommendations include more systematic field observations of C fluxes across
the boundaries of wetlands with the atmosphere, the upland and the river. Eddy
covariance data is still lacking in some remote wetlands where logistics are complicated
(Lu et al. 2016), for example in floodplains of large tropical rivers, which host highly
productive flooded forests and floating macrophytes (Piedade et al 1994; Morison et al.
2000), and largely contribute to riverine global $CO_2$ and $CH_4$ emissions (Richey et al.
2002; Engle et al. 2008; Bloom et al. 2010; Abril et al. 2014, Borges et al. 2015a). Eddy



covariance measurements should also be more systematically coupled at the same site
with chamber measurements, hydrological C fluxes and C sequestration studies but
accounting for the longer time-scale of the sequestration rates based on core dating.

The quantification in the field of the amount of C that enters or leaves wetland
ecosystems horizontally with water flow is challenging because many wetlands have
complex morphologies and multiple pathways of hydrological transport that should be
apprehended using hydrodynamical modelling. In addition to hydrological complexity,
the C forms may largely change when water crosses the wetland and for instance,
terrestrial mineral-bound POC can be trapped and replaced by wetland POC, DOC and
dissolved $CO_2$. Isotopic and molecular tracers can help in differentiating terrestrial from
wetland OC, when the signatures of the two sources are well separated, for instance, in
watersheds dominated by C3 forests, the contribution of wetland C4 macrophytes can
be tracked in riverine POC, DOC and DIC (Quay et al. 1992; Mortillaro et al. 2011; Albéric
et al. 2018). In contrast, OC from flooded forests is more difficult to differentiate from
that coming from *terra firme* forests (Ward et al. 2013) when many tree species are
common to both ecosystems (Junk et al. 2010). Radiocarbon age in rivers can be
interpreted as the time spent by C in soils and, when young C predominates, they
suggest a rapid transfer from plants to waters (Mayorga et al. 2005), as in wetlands
ecosystems. However, some wetlands such as peats can also export old dissolved C to
streams (Billet et al. 2007).

Original experimental work in mesocosms that simulate flooding, as well as wetland
ecosystem manipulations are necessary to characterize and quantify hydrological C
export per flooded area, as well as the fraction of ecosystem respiration occurring below





water; methods must be developed to estimate HR, ARw and ARs (Eq. 11-13). Soil core
incubations or submerged static chambers for instance, provide an estimate of HRs plus
a fraction of ARs in some wetlands with small plants; in the absence of phytoplankton,
dark bottle incubations measure HRw but miss ARw released by the submerged part of
plants. Special mesocosms adapted to the metabolism of semi-aquatic plants are
necessary. Data of metabolic rates are still missing but would be necessary to build
coupled hydrological-biogeochemical models for wetlands and inland waters. Process-
based biogeochemical models are indeed promising approaches for quantifying C
exports from flooded lands (e.g., Sharifi et al. 2013; Lauerwald et al. 2017). Ideally, these
models could simulate the most important biological processes in the wetland: GPP,
NPP, litter fall, and the different components of ER in air, water and soil, together with
hydrological transport and gas emission. Few modelling studies account for DOC export
(Sharifi et al. 2013), most miss the DIC export as dissolved $CO_2$ and do not correctly
account for the autotrophic respiration terms (ARw and ARs), or the heterotrophic
microbial processes in the root zone (HRs) (Fig. 2). Recently, Lauerwald et al. (2017)
developed a new type of model of C cycling in large rivers that mimics the most
important physical and biological processes, including land flooding; when applied to
the Amazon River, the model calculated a total $CO_2$ outgassing flux close to that upscaled
from field measurements (Richey et al. 2002); in addition, the computed relative
contributions to the total dissolved C inputs of surface runoff (14%), drainage (28%)
and flooding (57%) were consistent with recent field evidence that wetlands
predominantly fuel $CO_2$ outgassing from the Amazon River (Abril et al. 2014).

Finally, a precise upscaling of wetland and inland waters global C budgets requires an
adequate typology of C cycles that accounts for the different hydrological and



biogeochemical functioning of peats, swamps, marshes and floodplains, and their spatial
distributions along climatic zones (Lehner and Döll 2004). Ideally, process-based
models should be built and validated in individual wetland types, and then aggregated to
a global model able to quantify C fluxes between drained land, floodable land, rivers and
lakes and the atmosphere at the continental scale. Such modelling tools will also be
highly valuable to predict the impacts of climate and land use changes on these
continental C fluxes. Knowing the relative contribution of well-drained land and wetland
to inland water $CO_2$ emissions is crucial for quantifying the continental greenhouse gas
budget (Fig. 1) and to predict its sensitivity and feedback on climate warming. For
instance, the intensification of floods and droughts or river damming have the potential
to drastically modify C fluxes at the land-water-atmosphere interface and alter or
enhance the hotspot character of wetlands in the continental C cycle. Such evolution
must be monitored in the field, better understood, conceptualized, and modelled in
order to guide environmental conservation strategies in the next decades.

ACKNOWLEDGEMENTS
This contributes to the European Research Council Starting Grant AFRIVAL (240002),
the French national research agency CARBAMA project (grant n° 08-BLANC-0221), and
the CNP-Leyre project funded by the Cluster of Excellence COTE at the Université de
Bordeaux (ANR-10-LABX-45). AVB is a senior research associate at the Fonds National
de la Recherche Scientifique.





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



### a. Passive pipe

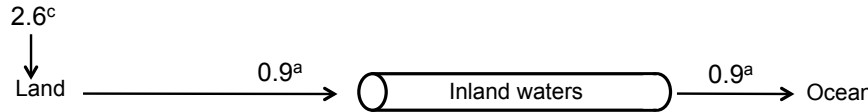

### b. Active pipe

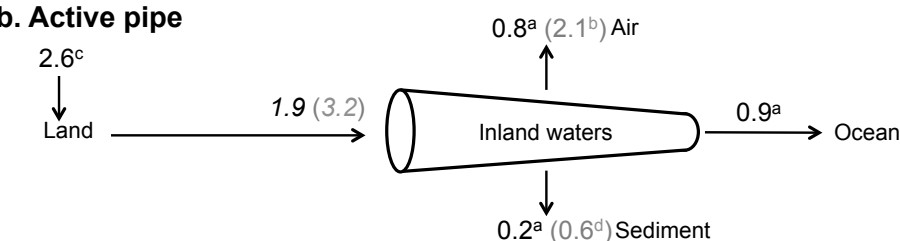

### c. Re-plumbed active pipe

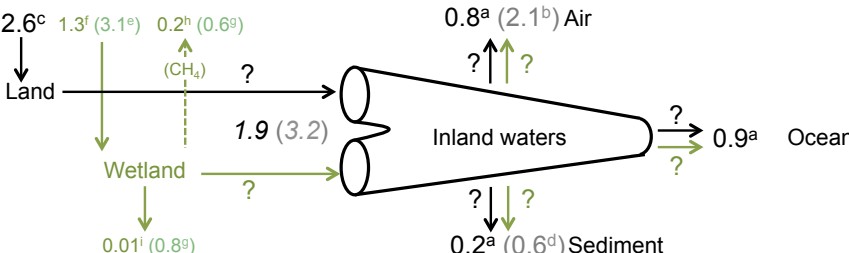


Fig. 1. An update of the active pipe concept, including wetlands in the C budget of inland
waters. [a] from Cole et al. (2007); [b] from Raymond et al. (2013) (note that the estimate of
global $CO_2$ outgassing from Cole et al. (2007) is similar to that of Lauerwald et al. 2015);
[c] calculated as the difference between land use change and net land flux in Ciais et al.
(2013); [d] from Tranvik et al. (2009); [e] from Lu et al. (2016); [f] from Lu et al. (2016)
corrected for a global wetland surface area of Downing et al. (2009); [g] from Mitsch et al.
(2013); [h] from Saunois et al. (2016); [i] corrected from Mitsch et al. (2013), according to
Bridgham et al. (2014). Numbers in italics are calculated as the sum of all others fluxes
and include a high (grey) and a low (black) estimate. Black arrows represent C
originating from well-drained, terrestrial ecosystems, and green arrows represent
wetland C.



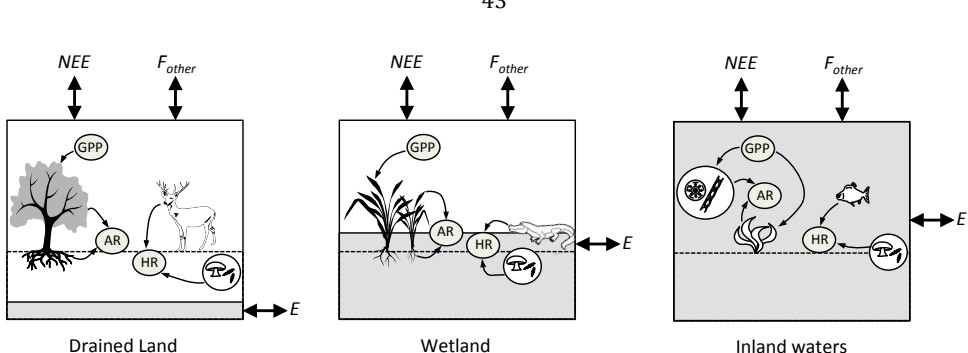


Fig. 2. Relationship among the carbon (C) fluxes (in italics) that determine net

ecosystem carbon balance (NECB) (the net of all C imports to and exports from the

ecosystem), and the metabolic fluxes (inside grey oval) that determine net ecosystem

production (NEP). (Adapted from Chapin et al. 2006 to include aquatic compartments).

The boxes represent the ecosystems (drained land, wetland, inland waters). Fluxes

contributing to NECB are (i) net ecosystem exchange (*NEE*) with the atmosphere

(emissions to or uptake from the atmosphere of carbon dioxide, $CO_2$); (ii) fluxes of

carbon forms other than $CO_2$ (*F_{other}*), which include methane ($CH_4$), carbon monoxide

(CO), and volatile organic C (VOC); (iii) lateral export (*E*) or import of dissolved organic

and inorganic C and particulate organic C by hydrological transport and other processes

such as animal movement, wind deposition and erosion, and anthropogenic transport or

harvest. In this study, we consider *F_{other}* as the flux of $CH_4$ from the ecosystem to the

atmosphere, and *E* as hydrological export from the ecosystem as POC, DOC, dissolved

$CO_2$ and dissolved $CH_4$. Fluxes contributing to NEP are gross primary production (GPP)

and ecosystem respiration (ER). ER includes autotrophic respiration (AR) by the

different components of vegetation (leaves, wood, roots and photosynthetic microbes)

and heterotrophic respiration (HR) by prokaryotes, fungi and animals. The shaded

volume in each box indicates the part of the ecosystem occupied by water. GPP and ER

occur mostly above the water table in well-drained ecosystems, partly above and below



the water table in wetland ecosystems, and exclusively in water and sediments in
aquatic ecosystems.



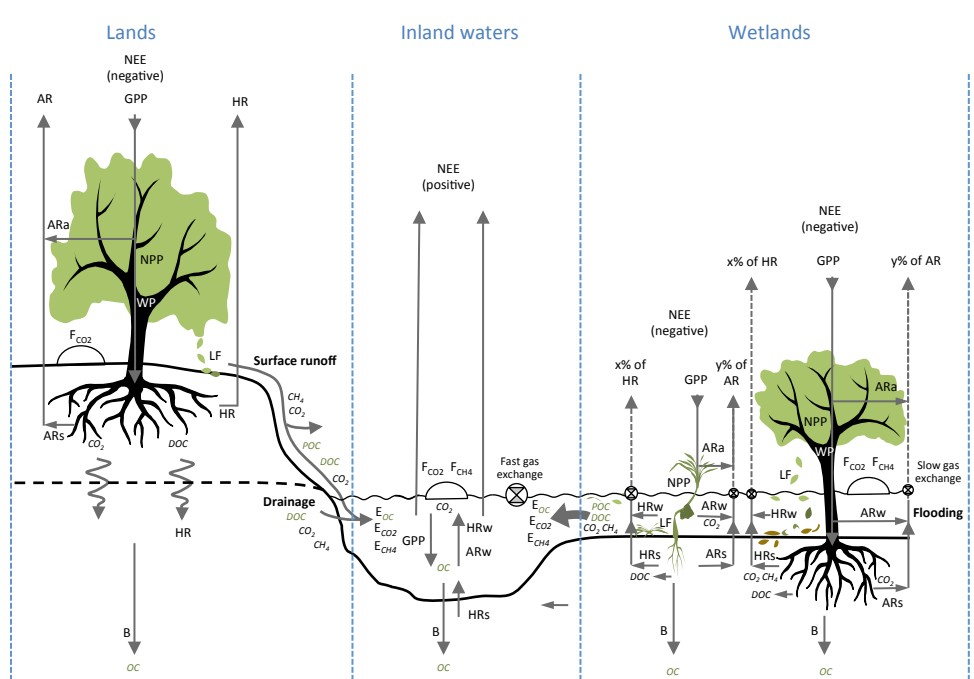


Fig. 3 Functional differences of carbon metabolism and hydrological export in well-drained and flooded land. NEE: net ecosystem exchange; GPP: gross primary production; NPP: net primary production; WP: wood production; LF: litter fall; AR: autotrophic respiration; ARa: autotrophic respiration in air; ARw: autotrophic respiration in water; ARs: autotrophic respiration in soils and sediments; HR: heterotrophic respiration; HRw heterotrophic respiration in water; HRs heterotrophic respiration in sediments; B: long-term burial in soils and sediments. POC: particulate organic C; DOC: dissolved organic C; $E_{OC}$: export of organic carbon (sum of DOC and POC); $E_{CO2}$: export of dissolved $CO_2$; $E_{CH4}$: export of dissolved $CH_4$; $F_{CO2}$ and $F_{CH4}$: fluxes of $CO_2$ and $CH_4$ at the soil-air or water-air interface (as determined with static chambers). Note that, by convention, NEE is opposite in sign to GPP and NPP because NEE is defined by atmospheric scientists as a C input to the atmosphere, whereas GPP and NPP are defined by ecologists as C inputs to ecosystems (Chapin et al. 2006). C export to river systems results from the interactions between metabolic processes and C transport processes between air, plants, soils,



sediments and waters and are fairly different in wetlands ecosystems (right) and
terrestrial, never-flooded, ecosystems (left). In terrestrial systems, carbon export occurs
as surface runoff and drainage and includes a small fraction of LF, root exudation, ARs,
and HR. In contrast, in flooded wetlands (right), almost all LF and root exudation (that
releases DOC), as well as a substantial fraction of ecosystem respiration
(ARw+ARs+HRw+HRs) are transferring C to the aquatic system as OC and dissolved
gases; in addition, slow gas exchange (low gas transfer velocity) in protected wetlands
favours lateral export of dissolved $CO_2$ and $CH_4$. These lateral C fluxes are enhanced in
wetlands compared to drained systems and should generate strong discrepancies
between ecosystem metabolic fluxes (GPP, NPP, ER, and NECB) and vertical C fluxes
measured in the field with static chambers ($F_{CO2}$ and $F_{CH4}$), and eddy covariance towers
(NEE).
