# Peer review of "Carbon leaks from flooded land: do we need to re-plumb the inland water active"

_Biogeosciences, 2018_

## Referee Comment (RC1) · Anonymous Referee #1 · 5 Jul 2018

The manuscript by Abril and Borges discusses existing conceptions of inland waters in the global C cycle and presents an updated view with a stronger focus on inland water-wetland interactions. While traditional conceptions see upland terrestrial ecosystems as only allochtonous source of C to inland waters, wetlands are known to be an important source of C to inland waters while having a specific ecology which is distinct from both terrestrial ecosystems and inland waters. This new conception is timely as it finally allows for a more complete perception of C cycling through the terrestrial-aquatic continuum of the continental surface.

Both authors have a great international reputation in the field of inland water and wet-

land biogeochemistry, and their own work has in the past largely contributed to the growing awareness of the importance of wetland-inland water interactions for the biogeochemistry of inland waters. Their long-standing expertise becomes quite apparent in the presented manuscript. The review of existing literature in the field is very complete and their own ideas and perspectives are clearly described in a comprehensive and logically sound manner. I am sure that this manuscript will be of great interest for the readership of Biogeochemistry, and I recommend publication after minor revisions.

**General comments**

L38-39: "primary production and respiration in air" What do you mean by "in air"? Above ground/water table?

L59-60: You need a reference for that.

L73-78: Here you should quickly mention that reservoirs are an important form of man-made inland waters.

L90-91: However, Garrels and Mackenzie 1971 were also among the first to show the general CO2 oversaturation in rivers.

Eq. 1: E and Fother should be net fluxes, as ecosystem can for instance take up atmospheric CH4 and as there can also be lateral imports from upstream.

L146-148: Does this exclude or include weathering related fluxes of DIC? Please, clarify.

L189-191: Here, make clear that the weathering of carbonate rocks also involves a mineral source of DIC. That is trivial, but may not be that obvious to the broad readership.

L244-248: Lauerwald et al. used a 0.5° x 0.5° grid

L291-293: I don't think that Krinner et al. 2005 is an adequate reference here. That's the paper describing the standard version of ORCHIDEE which does not include fluvial C fluxes. Only very recently, models have been developed which include fluvial C fluxes: e.g. DLEM (Tian et al., 2015) and ORCHILEAK (Lauerwald et al., 2017). JULES-DOCM (Nakhavali et al., 2018) is a land surface model that accounts at least for the leaching of DOC from soils.

L416-425: Here I find it a bit odd to report "-NEE", and not just NEE with their negative values. But that's maybe a question of taste.

L450-451: I think there is a word missing in that sentence.

Eq. 13: You should define the meaning of $\beta$, like "fraction exported laterally", or something similar. It's obvious from the equations, but it would be nice to have it written in words.

L550: What do you mean by "community"? An ecological community, i.e. the assembly of organisms in one ecosystem?

---

## Referee Comment (RC2) · Anonymous Referee #2 · 12 Oct 2018

This paper is a timely contribution to the discussion about the role of inland waters in the global C cycle in that it connects two important aquatic elements (emergent wetlands and rivers/lakes). I found the paper to be provocative, rigorous and insightful, and thus look forward to its publication. I have several comments on the paper that are mostly second-tier issues (i.e., none challenge the core arguments, just more minor details of those arguments), as well as several editorial suggestions.

1) The equations provided are useful, but there a a few issues that the authors could consider to augment. Principal among these was the utility of a master equation that connects Eq. 1 and Eq. 2. The text is full of compelling subleties about where NEE

departs from NEP, and where NEP departs from NECB, and these are central to the overall argument. I think that returning to this master equation for each section (aquatic, terrestrial and wetland) would integrate the narrative more clearly. When the equations for the wetland budgets are presented, new terms (alpha, beta) are introduced. Alpha is described, but beta is not directly (i.e., proportion of aquatic CO2 that is transported laterally). The general use or not of subscripts to denote fractions (e.g., E[subCO2]) is not general (e.g. ARw and ARs).

2) The authors do a really nice job integrating inorganic C and organic C into the narrative. That said, I feel like there is a missed opportunity in the wetland and stream sections to actually enumerate the relative importance of the two modes of C transport. In the area where I work, DOC overwhelmingly dominates mass transport (mean DOC $\sim$ 40 mg C/L, mean excess DIC $\sim$ 3 mg C /L), suggesting a slightly different emphasis than the current paper takes (for which the focus is mostly on DIC transport). One place this manifests as an issue is L441-443. There the authors assert that if E is large (in this case not parsing organic and inorganic species), then GPP, NPP and NEP will be overestimated and ER underestimated. This is ONLY true if E is principally DIC. If, however, most of the lateral flux is DOC then the opposite would be true.

3) Throughout the paper, the authors assume strong hydrologic connectivity between wetlands and other inland waters. This is true, at least episodically, for many riparian wetlands, particularly those along large rivers. This raises an important issue, however. Many many wetlands are NOT well connected, but rather venues of distributed water storage that connect only for short intense periods, but where otherwise water flowpaths to convey C are neither rapid nor volumetrically significant. This storage is, indeed, the main reason that wetlands provide flood attenuation services. While I cannot claim experience with wetlands in all areas, the wetlands that I do work in are mostly disconnected expect via slow groundwater flowpaths except during short bursts of event driven connectivity. As such, the C budget in those flooded lands is mostly entirely vertical (during periods of weak connectivity) except when connectivity enables

transport. I think this means that for many of the wetlands that I know well, the lateral fluxes are likely to be very small for most hydrologic conditions.

4) Building on #3, the time variation of lateral fluxes and respiration pathways is not clearly considered. On P24, the authors assert that nearly all of ARs and HR occur sub-aqueous. This is not the case, unless one assumes that wetlands are permanently inundated. Most wetlands are not permanently inundated, and have sometimes pro-longed periods when soils are no longer saturated, during which there is frequently a significant pulse of respiration with the atmosphere serving as the destination. This is both ARs and HR. I admit that this is a relatively minor nuance, and does not change the overarching assertion, but measuring reasonable values of alpha and beta (not to mention ARs and HR) requires that our conceptual model remain faithful to the ac-tual processes. The timing issues are important not just because they apply to nearly every "plumbing" flowpath, but also because time variation in concentrations is infor-mative about sources. In particular, the fact that DIC, DOC and POC all generally increase with increasing discharge suggests sources that are activated during events (consistent with the wetland source narrative). If the source were principally terrestrial, it seems to me that dilution signals would be more frequently observed.

5) The thought experiment on P24 and 25 to estimate the proportion of respired $CO_2$ that is exported has some flaws. First, the gas transfer velocity is way too high; 1 cm s-1 is about 3000 times larger than values that we measure. I checked the Ho et al. (2018) reference and their gas transfer velocities are 1 cm hr-1, which aligns with measure-ments we've made, and likely represents a unit error. This should greatly INCREASE the fraction of $CO_2$ that is laterally exported, except that the velocities that are used are (at least for the wetlands that I work on) way too high. As I mentioned above, velocities are typically $\sim$0 except during event driven connectivity periods (neglecting the very modest, but non-zero, velocities associated with groundwater transport). Even the lotic wetlands have maximum velocities of 1 cm s-1. These two assumptions cancel out, possibly rendering the ultimate inference sound. I'd add that even if the beta values are

far lower (say 5%), this is still a significant mass flux. One convenient way to frame this argument is using the Damkohler number relating advection to reaction (in this case gas exchange). I could imagine a typological synthesis across wetlands that highlights very low Damkohler numbers for distributed depressional storage, and higher numbers for riparian and littoral wetlands.

A few more minor comments:

- As I mentioned above, wetlands do not always exhibit "strong hydrological connectivity" (L27).

- I am not convinced that the authors have sufficiently demonstrated the primacy of wetland C loading in the tropics. The mechanisms are general, and the absence of direct data to support that the role of wetlands is disproportionately important in the tropics makes inclusion of that conclusion in the abstract a little bit of a stretch.

- phrasing on L168/169 is very odd. "intermittent and/or vegetated flooded land" makes no sense to me as a wetland scientist.

- The structure of the text on P8-9 yields a "Second,..." on L176 that it took some hunting to find the accompanying "first". Consider revising.

- L199 -is there a citation for this?

- I feel as though the emphasis in L286-287 is so much on DIC that DOC fluxes are getting lost. DOC fluxes can be significant in numerous low-relief settings, even from uplands.

- The mass budget closure of terrestrial systems was pretty compelling, but the conviction of the narrative on L321-323 seemed strong given how close those numbers get. The NEE vs. NECB difference could plausibly be as large as 0.7 Pg yr-1, which is in the range of OC burial + export rates.

- L396 - extra "has"

- L474 - I didn't understand the "Albeit..." part of this sentence.

- L483 - Plants transporting gases from the soil to the air (and vice versa) is NOT why wetlands are generally hypoxic. They are hypoxic because the gas exchange of oxygen is slow compared to consumption. I wonder if there was another sentence there in an earlier draft.

- L504 - should be Mitsch et al. (2013), or another reference that is missing from the list.

- The controls on beta are as described (depth, gas exchange) but also critically hydrological connectivity (which is not a constant property).

- Fig. 3 is great. It's worth noting that the wetland depicted is one modest subset of wetland area (i.e., those with perennial connections to other inland waters).

---

## Author Response (AR1)

Gwenaël Abril Directeur de Recherche au CNRS UMR BOREA (Biologie des Organismes et Ecosystèmes Aquatiques) MNHN- CNRS 7208 - IRD 207- UPMC- UCN- UA 61 rue Buffon, 75231, Paris cedex 05, France.

Dear Dr Battin,

Thank you very much for considering our submitted paper with minor revisions. We have now revised our MS in order to address most of the comments by the two reviewers; these comments, particularly those of reviewer #2, led us to make important modification in the text (particularly in sections 2, 3, 5 and 6), that you can follow in the marked version below. Basically, changes concerned the addition of information on surface areas by climatic regions (new Table 1) in order to strengthen our reasoning about the importance of tropical wetlands, some discussion about wetlands with little connectivity with rivers or lakes, such as peat bogs and swamps (section 5), and about temporal variation of C export by flooding. The question of hydrological connectivity of wetlands has been intensively addressed. Through all the text and in figures 2 and 3, we also changed the term "wetland" to "flooded land" when appropriate. Thanks to this very constructive round of review, we believe our revised paper convincingly describes the inconsistencies between studies from terrestrial and aquatic perspectives, and demonstrates that these inconsistencies can be solved by considering land flooding as a major mechanism for the export of terrestrial C to rivers, together with runoff and drainage. We suggest the occurrence of a wetland CO2 pump that efficiently transfer atmospheric carbon to river systems, and propose an update of the renowned "plumbing paper" by Cole and coll. in 2007. Finally, we provide so guidelines in order to fill the important gaps in scientific knowledge of carbon fluxes at the land-water interface. Because our paper contains cutting-edge ideas about the continental carbon cycle and original concepts on aquatic ecology and biogeochemistry we believe it will be of great interest to a large audience and could appear as highlight article on the BG web site.

1

With best regards Gwenaël Abril

Niterói, 29/12/2018
COMMENT- The manuscript by Abril and Borges discusses existing conceptions of inland waters in the global C cycle and presents an updated view with a stronger focus on inland waterwetland interactions. While traditional conceptions see upland terrestrial ecosystems as only allochtonous source of C to inland waters, wetlands are known to be an important source of C to inland waters while having a specific ecology which is distinct from both terrestrial ecosystems and inland waters. This new conception is timely as it finally allows for a more complete perception of C cycling through the terrestrial-aquatic continuum of the continental surface.

Both authors have a great international reputation in the field of inland water and wetland biogeochemistry, and their own work has in the past largely contributed to the growing awareness of the importance of wetland-inland water interactions for the biogeochemistry of inland waters. Their long-standing expertise becomes quite apparent in the presented manuscript. The review of existing literature in the field is very complete and their own ideas and perspectives are clearly described in a comprehensive and logically sound manner. I am sure that this manuscript will be of great interest for the readership of Biogeochemistry, and I recommend publication after minor revisions.

ANSWER- We thank the reviewer for her/his encouraging comments and positive evaluation of our MS.

**General comments**

COMMENT- L38-39: "primary production and respiration in air" What do you mean by "in air"? Above ground/water table?

ANSWER- We will change the text to "submerged and emerged parts of the plants"

COMMENT- L59-60: You need a reference for that. ANSWER- We will cite Ciais et al. 2013 here.

COMMENT- L73-78: Here you should quickly mention that reservoirs are an important form of man- made inland waters.

ANSWER- We will write here "streams, lakes, reservoirs, rivers and estuaries"

COMMENT- L90-91: However, Garrels and Mackenzie 1971 were also among the first to show the general CO2 oversaturation in rivers.

ANSWER- Indeed, Garrels and Mackenzie (1971) computed pCO2 in a few large rivers based on pH and HCO3- data from Durum et al. (1960), using a simplified and approximate computation scheme. They mention briefly (in the legend of the figure) that the pCO2 values in rivers are above atmospheric equilibrium. However, they did not estimate the emission of CO2 to the atmosphere from global rivers, not put this number into perspective with regards to other components of the carbon cycle such as organic carbon transport to the ocean. This was only done from the 1990's onwards, first with paper of Cole et al. (1994) in lakes, then the Cole & Caraco (2001) paper on rivers that were then synthesized with additional data in the Cole et al. (2007) paper.

COMMENT- Eq. 1: E and Fother should be net fluxes, as ecosystem can for instance take up atmospheric CH4 and as there can also be lateral imports from upstream.

ANSWER- Yes indeed. In fact all fluxes are net fluxes (positive and negative) including also NEE, NECB. We will specify this in the text

COMMENT- L146-148: Does this exclude or include weathering related fluxes of DIC? Please, clarify.

ANSWER- Indeed, this deserves a clarification, because DIC from carbonate rock should be excluded from this statement. New phrasing will be: "One process that makes -NEE diverge from NEP and NECB is when significant amounts of inorganic C enter or leave the ecosystem as DIC in the aquatic phase with horizontal hydrological transport rather than through atmospheric exchange (Chapin et al. 2006). However, DIC from mineral source of carbonate rock weathering will not contribute to the difference between NEP and NECB. In addition to this divergence between NEE and NEP, NECB deviates from NEP when C enters or leaves the ecosystem in forms others than CO2 or DIC (Eq. 1). This includes horizontal transport of particulate and dissolved OC..."

COMMENT- L189-191: Here, make clear that the weathering of carbonate rocks also involves a mineral source of DIC. That is trivial, but may not be that obvious to the broad readership.

ANSWER- New phrasing will be "Weathering of carbonate and silicate rocks is mediated by soil  $CO_2$  derived from respiration, so that weathering is also a component of ER; however, the weathering of carbonate rock involves another mineral source of DIC which contributes to half of the alkalinity produced".

COMMENT- L244-248: Lauerwald et al. used a 0.5 grid

ANSWER- Corrected in the revised version

COMMENT- L291-293: I don't think that Krinner et al. 2005 is an adequate reference here. That's the paper describing the standard version of ORCHIDEE which does not include fluvial C fluxes. Only very recently, models have been developed which include fluvial C fluxes: e.g. DLEM (Tian et al., 2015) and ORCHILEAK (Lauerwald et al., 2017).

JULES-DOCM (Nakhavali et al., 2018) is a land surface model that accounts at least

for the leaching of DOC from soils.

ANSWER- In the revised MS, we will remove the Krinner et al. 2005 ref and add the Tian et al., 2015 Lauerwald et al., 2017 and Nakhavali et al., 2018 references.

COMMENT- L416-425: Here I find it a bit odd to report "-NEE", and not just NEE with their negative values. But that's maybe a question of taste.

ANSWER- We find easier and clearer to use "-NEE" because it avoids any confusion in the phrasing when comparing with NEP and NECB: "-NEE is higher than NEP" is easy to understand.

**COMMENT- L450-451: I think there is a word missing in that sentence.**

ANSWER- New sentence reads: "One problem with NPP data is that it does not account for all the C transferred by the plants from the atmosphere to the soil and water."

COMMENT- Eq. 13: You should define the meaning of, like "fraction exported laterally", or something similar. It's obvious from the equations, but it would be nice to have it written in words.

ANSWER- we define  $\beta$  in the revised MS, and we will also write " $\alpha\beta$  is thus the fraction of ecosystem respiration that is exported laterally from the wetland in water masses."

COMMENT- L550: What do you mean by "community"? An ecological community, i.e. the assembly of organisms in one ecosystem?

ANSWER- we remove the ambiguous term "community"

Anonymous Referee #2

Received and published: 12 October 2018

COMMENT- This paper is a timely contribution to the discussion about the role of inland waters in the global C cycle in that it connects two important aquatic elements (emergent wetlands and rivers/lakes). I found the paper to be provocative, rigorous and insightful, and thus look forward to its publication. I have several comments on the paper that are mostly second-tier issues (i.e., none challenge the core arguments, just more minor details of those arguments), as well as several editorial suggestions.

ANSWER- We thank the reviewer for her/his interest and detailed reading of our MS, her/his overall positive evaluation of our paper and encouraging and constructive comments. The reviewer has raised very relevant points and asked important questions concerning our understanding of carbon exchange between wetlands and inland waters; on several aspects, the reviewer highlighted some limits in our reasoning, particularly when considering wetlands only when connected with inland waters. Owing to the general limited knowledge on these topics, it is obvious we could not provide a definitive answer to all his/her five major comments; this is why we acknowledge the reviewer when he/she recognises our contribution insightful and rigorously and when she/he mentions that his/her comments are second-tier issues. Nevertheless, in our revised MS, we will temple some of our statements in the light of these comments, give more evidence on the mentioned facts based on new references to literature, put more emphasis on the remaining uncertainties, and we provide some additional quantitative information useful to support our views, particularly those on the relative importance of tropical regions. These changes will improve our MS by giving more emphasis on current gaps and the necessity to fill the gaps between land and water in the future.

COMMENT- 1) The equations provided are useful, but there a a few issues that the authors could consider to augment. Principal among these was the utility of a master equation that connects Eq. 1 and Eq. 2. The text is full of compelling subtleties about where NEE departs from NEP, and where NEP departs from NECB, and these are central to the overall argument. I think that returning to this master equation for each section (aquatic, terrestrial and wetland) would integrate the narrative more clearly.

ANSWER: we will do our best to follow this excellent but very challenging suggestion; please note that in fact, no single universal equation connects Eq.1 and Eq.2, exactly because in almost all cases, all terms are significant and NEE departs from NEP and from NECB. However, it is indeed possible to connect Eq. 1 and Eq.2 if the objective is to illustrate some assumptions commonly made by ecologists, and the consequences of these assumptions on C fluxes conceptualization.

In aquatic systems, the general case is a positive NEE (CO2 source), a negative NEP (heterotrophy), a negative E (import from surrounding land) NECB can be positive (if burial of terrestrial C exceeds heterotrophy) or negative and and  $F_{other}$  can be reasonably neglected if CH4 emissions from open waters only are considered (L222-223); thus Eq.1 and Eq.2 can be combined to: -NEE = NEP +  $E_{CO2}$  Consistent with the term "internal" (-NEP) and "external CO2" (-E).

We will insert this simplified equation together with few explicative sentences in the aquatic section after L286.

"In inland waters, Eq.1 and Eq.2 are generally combined to a simplified equation that only considers inorganic C :

 $-NEE = NEP + E_{CO2}$  Eq. X

with NEE positive, NEP negative (heterotrophic metabolism), and  $E_{CO2}$  negative, as rivers and lakes receive more dissolved  $CO_2$  from groundwater than they export downstream."

In terrestrial systems, the general approach based for instance on eddy-covariance CO2 fluxes often consists in neglecting  $F_{other}$  and E; thus Eq.1 and Eq.2 are combined to:

-NEE=NECB=NEP=GPP-ER; we will insert this statement in the terrestrial section L370:

..."by considering the loss of CO2 that dissolves in groundwater as negligible or within the error of estimation of metabolic flux at the ecosystem scale. In other terms, classical approaches in terrestrial ecosystems consisted in neglecting Fother and E and combine Eq.1 and Eq. 2 to: -NEE=NECB=NEP=GPP-ER Eq. Y"

In flooded systems, similar hypothesis cannot be made to link Eq.1 with Eq.2. In fact the relationship between fluxes in and out of the wetland (Eq.1) are linked to metabolic fluxes (Eq.2) according to the series of Eq. 14 to 16. Eq. 16 is the correct equation processes in the flooded area atmospheric exchange and is more complex for many reasons

**NPP+ $\beta$ ARw+ $\beta$ ARs- $(1-\beta)$ HR =B+FC02+FCH4+E**

In the revised MS, we will modify the last sentence of the paragraph L546-547 "*The three terms ARw and ARs and HR together with the E term, are generally neglected in wetland C budgets* that quantify only NPP, FCO2, FCH4 and B (*Mitsch et al. 2013; Sjögersten et al. 2014*)."

COMMENT- When the equations for the wetland budgets are presented, new terms (alpha, beta) are introduced. Alpha is described, but beta is not directly (i.e., proportion of aquatic CO2 that is transported laterally). The general use or not of subscripts to denote fractions (e.g., E[subCO2]) is not general (e.g. ARw and ARs).

ANSWER- We will define beta in the text and use subscripts for all terms of the equations

COMMENT - 2) The authors do a really nice job integrating inorganic C and organic C into the narrative. That said, I feel like there is a missed opportunity in the wetland and stream sections to actually enumerate the relative importance of the two modes of C transport. In the

area where I work, DOC overwhelmingly dominates mass transport (mean DOC 40 mg C/L, mean excess DIC 3 mg C /L), suggesting a slightly different emphasis than the current paper takes (for which the focus is mostly on DIC transport). One place this manifests as an issue is L441-443. There the authors assert that if E is large (in this case not parsing organic and inorganic species), then GPP, NPP and NEP will be overestimated and ER underestimated. This is ONLY true if E is principally DIC. If,

however, most of the lateral flux is DOC then the opposite would be true.

ANSWER- we recognize that we have put more emphasis on inorganic lateral fluxes rather than organic fluxes, probably because of our own experience in tropical floodplain systems, where export of excess DIC apparently predominates. In our revised MS, we will put more emphasis on the fact that some wetlands like peat bogs for instance exports DOC rather than DIC, citing:

Freeman C., Evans C.D. and Monteith (2001) Export of organic carbon from peat soils. Nature 412: 785

Clark J.M., Lane S.N., Chapman P.J. and Adamson J.K. (2008) Link between DOC in near surface peat and stream water in an upland catchment. Science of the total environment 404: 308-315

The sentence L441-443 was: "if wetland E is ignored but significant, GPP, NPP, NEP, and NECB deduced from the diurnal changes of eddy CO2 fluxes (Lu et al. 2016) would be overestimated and, inversely, ER would be underestimated (Eqs.1-6)". We agree with the referee that this sentence deserves more attention and must be moderated in some way. With the eddy covariance technique, NEE is measured throughout day and night; ER is assumed equal to nighttime NEE, ER is correlated to temperature; GPP is calculated as daytime NEE minus ER recalculated for daytime temperature; NPP is calculated from the diurnal integration of GPP minus ER. Thus, in the revised MS, we will first limit our reasoning to GPP, NPP and ER. If the wetland exports DIC, then this DIC will be missing in the night-time positive NEE and its relationship with temperature that are used to calculate ER; thus ER will be underestimated, and GPP and NPP will be overestimated. If the wetland exports DOC, this organic carbon will not be respired within the ecosystem and thus the assumption that ER equals nighttime NEE will remain valid. Finally, we agree with the reviewer's comment that "this is ONLY true if E is principally DIC", but we disagree with the statement that "if however, most of the lateral flux is DOC then the opposite would be true" because in theory, DOC export would have no effect ER, GPP and NPP deduced from eddy covariance; If DOC export is significant, NECB will be overestimated; however, NECB cannot be derived directly from the eddy covariance method (reference to Lu et al. 2016). In the revised MS, we will specify that the sentence is true if E is principally DIC and remove "NECB" from the sentence above.

COMMENT- 3) Throughout the paper, the authors assume strong hydrologic connectivity between wetlands and other inland waters. This is true, at least episodically, for many riparian wetlands, particularly those along large rivers. This raises an important issue, however. Many many wetlands are NOT well connected, but rather venues of distributed water storage that connect only for short intense periods, but where otherwise water flowpaths to convey C are neither rapid nor volumetrically significant. This storage is, indeed, the main reason that wetlands provide flood attenuation services. While I cannot claim experience with wetlands in all areas, the wetlands that I do work in are mostly disconnected expect via slow groundwater flowpaths except during short bursts of event driven connectivity. As such, the C budget in those flooded lands is mostly entirely vertical (during periods of weak connectivity) except when connectivity enables transport. I think this means that for many of the wetlands that I know well, the lateral fluxes are likely to be very small for most hydrologic conditions.

ANSWER- it is true that many wetlands are only temporally connected to inland waters; as a corollary, only wetlands strongly connected with inland waters will contribute disproportionally to the inland water C budget. We will make this point clearer in our revised MS by inserting the appropriate statements in sections 2 and 5. Because of the complexity of processes and the

spatial and temporal variability of wetland ecosystem functioning, in order to apprehend land flooding as mechanism for lateral C transport, we use land flooding as operational criteria to delimit continental areas (based for instance on remote sensing) as a major process that contribute to the wetland to inland water C flux. Thus, as the reviewer emphases, wetlands almost permanently flooded will contribute continuously, whereas wetlands episodically flooded will contribute only during short periods, although C lateral fluxes during these short periods can still be significant in the annual C budget.

Two major reasons justify to put more emphasis throughout our MS on riparian and littoral wetlands: firstly, these type of wetlands directly connected to river and lake water bodies will exchange more C with inland water; secondly, these types of wetlands predominate in tropical regions owing to the specific hydrological features (dry and rainy seasons), and according to Raymond et al. (2013) and Lauerwald et al. (2015) almost 80% of CO2 emissions from inland waters occur at latitude lower than 25°. In sections 2 and 5 of the revised MS, we will put more emphasis on the difference between swamps, bogs and marshes on the one hand and riparian and littoral wetlands on the other hand. A special paragraph will appear in the section 6 ("what tools do plumbers need?") on the absolute necessity to build a global typology of wetlands that adequately address the question of hydrological connectivity with inland waters.

COMMENT- 4) Building on #3, the time variation of lateral fluxes and respiration pathways is not clearly considered. On P24, the authors assert that nearly all of ARs and HR occur subaqueous. This is not the case, unless one assumes that wetlands are permanently inundated. Most wetlands are not permanently inundated, and have sometimes prolonged periods when soils are no longer saturated, during which there is frequently a significant pulse of respiration with the atmosphere serving as the destination. This is both ARs and HR. I admit that this is a relatively minor nuance, and does not change the overarching assertion, but measuring reasonable values of alpha and beta (not to mention ARs and HR) requires that our conceptual model remain faithful to the actual processes. The timing issues are important not just because they apply to nearly every "plumbing" flowpath, but also because time variation in concentrations is informative about sources. In particular, the fact that DIC, DOC and POC all generally increase with increasing discharge suggests sources that are activated during events (consistent with the wetland source narrative). If the source were principally terrestrial, it seems to me that dilution signals would be more frequently observed.

ANSWER- Yes, we agree that our reasoning do not explicitly consider wetland C balance during prolonged emersion; indeed, during these periods, C fluxes are mostly vertical, with lateral C fluxes occurring only as subterranean flow, similarly to drained land. Lateral fluxes induced by flooding follow seasonal cycles and can occur as regular flood pulse such as in many tropical wetlands, or as flash flood events. The proportion of wetland GPP exported annually to streams and rivers can be highly significant in both modes; however it would make sense if regular flooding export mostly DIC whereas flash flood export mostly DOC and POC. Probably because remote sensing tools permit their clear delimitation in space and time, aquatic scientists use to consider wetlands as the flooded area only, even if this area changes with time. This is why we preferred to use "flooded land" and not "wetland" in the title of the MS. We do not neglect the emerged period of wetland, but we consider it belongs to the terrestrial domain. We believe we can assume this limitation because our main focus is the description of the C transport mechanism induced by flooding, and not the net C balance of wetland ecosystems as ecological entities. We agree with the reviewer that we must make it very clear in the MS, particularly in sections 2, 5 and 6. In the revised MS, more emphasis will be given to temporal variation of flooding and export. Section 6 ("plumber tools") will contain a short paragraph on the importance of building a global typology of wetlands that include hydrological connectivity and other ecosystems specificities such as productivity, CH4 fluxes, etc... In order to mitigate this defect in our MS, we will also change "wetlands" to "flooded land" when appropriate.

We thank the reviewer for sharing constructive ideas on the analysis of concentration versus discharge patterns. We agree with the idea that positive slopes suggest activation of a wetland source and a larger export of wetland C at high discharge.

COMMENT-5) The thought experiment on P24 and 25 to estimate the proportion of respired CO2 that is exported has some flaws. First, the gas transfer velocity is way too high; 1 cm s-1 is about 3000 times larger than values that we measure. I checked the Ho et al. (2018) reference and their gas transfer velocities are 1 cm hr-1, which aligns with measurements we've made, and likely represents a unit error. This should greatly INCREASE the fraction of CO2 that is laterally exported, except that the velocities that are used are (at least for the wetlands that I work on) way too high. As I mentioned above, velocities are typically 0 except during event driven connectivity periods (neglecting the very modest, but non-zero, velocities of 1 cm s-1. These two assumptions cancel out, possibly rendering the ultimate inference sound. I'd add that even if the beta values are far lower (say 5%), this is still a significant mass flux. One convenient way to frame this argument is using the Damkohler number relating advection to reaction (in this case gas exchange). I could imagine a typological synthesis across wetlands that highlights very low Damkohler numbers for distributed depressional storage, and higher numbers for riparian and littoral wetlands.

ANSWER- we apology for the mistake in the unit. In our calculation, we have used a gas transfer velocity of 1 cm hr-1 and not 1 cm s-1. As mentioned above in previous answers, the focus of our paper is to discuss land flooding as a process that transport C to inland waters, and this is relevant only when flooding occurs and when surface water velocities within wetlands are non-zero. A water velocity of 1 cm s-1 is indeed classical for lotic wetland. However, in floodplains of large tropical rivers which are significant contributors of tropical and global CO2 and CH4 budget, water velocities inside a flooded forest can reach several dozens of cm s-1 during the months of maximum flood, when dissolved CO2 concentrations are maximum. 10 cm s-1 can be assumed as a maximum velocity value. This observation also contributes to highlight the importance of temporal changes in the lateral fluxes between wetlands and rivers (comment 4 and corresponding answer).

**A few more minor comments:**

COMMENT- As I mentioned above, wetlands do not always exhibit "strong hydrological connectivity" (L27).

ANSWER- the sentence will be changed to "Contrarily to well-drained land, many wetlands combine a strong and dynamic hydrological connectivity with inland waters, high productivity..."

COMMENT - I am not convinced that the authors have sufficiently demonstrated the primacy of wetland C loading in the tropics. The mechanisms are general, and the absence of direct data to support that the role of wetlands is disproportionately important in the tropics makes inclusion of that conclusion in the abstract a little bit of a stretch.

ANSWER- we agree with reviewer that the mechanisms we describe are general; however, their quantitative significance at the global scale is probably much more important in the tropic, where 80% of global river CO2 degassing occurs. We will mention this fact in the abstract.

COMMENT- phrasing on L168/169 is very odd. "intermittent and/or vegetated flooded land" makes no sense to me as a wetland scientist.

ANSWER- we will remove this definition probably useless for our main message.

COMMENT- The structure of the text on P8-9 yields a "Second,..." on L176 that it took some hunting to find the accompanying "first". Consider revising.

ANSWER- We will write L153 "As a first step, an adequate conceptualization..." and L176 "As a second step, our conceptual model should be two-dimensional..."

COMMENT- L199 -is there a citation for this? ANSWER- We will cite Chanton et al. (1995) here

COMMENT- I feel as though the emphasis in L286-287 is so much on DIC that DOC fluxes are getting lost. DOC fluxes can be significant in numerous low-relief settings, even from uplands.

ANSWER- We will put more emphasis on the importance of DOC export from some wetlands such as peats citing Freeman et al. (2001) and Clark et al. (2008), in this paragraph of the MS as well as in the latest section 6.

COMMENT - The mass budget closure of terrestrial systems was pretty compelling, but the conviction of the narrative on L321-323 seemed strong given how close those numbers get. The NEE vs. NECB difference could plausibly be as large as 0.7 Pg yr-1, which is in the range of OC burial + export rates.

ANSWER- Indeed, we will modify this sentence to state that difference between NEE and NECB is still is the range of OC burial+export.

COMMENT- L396 - extra "has". - L474 - I didn't understand the "Albeit..." part of this sentence. ANSWER- Will be re-phrased in the revised MS

COMMENT - L483 - Plants transporting gases from the soil to the air (and vice versa) is NOT why wetlands are generally hypoxic. They are hypoxic because the gas exchange of oxygen is slow compared to consumption. I wonder if there was another sentence there in an earlier draft. ANSWER- We agree with the reviewer. In the revised MS, will remove the words "This is why"

COMMENT- L504 - should be Mitsch et al. (2013), or another reference that is missing from the list.

ANSWER- Indeed, the reference is Mitsch et al. (2013)

COMMENT - The controls on beta are as described (depth, gas exchange) but also critically hydrological connectivity (which is not a constant property).

ANSWER- in our beta is also a function of water velocity, that is of hydrological connectivity; we will mention that in the revised MS.

COMMENT- Fig. 3 is great. It's worth noting that the wetland depicted is one modest subset of wetland area (i.e., those with perennial connections to other inland waters).

ANSWER- We thank the reviewer for the compliment. We will change the title of the subsets delimiting ecosystems type; change "land" to "drained land" and "wetland" to "flooded land". We will mention in the legend that with such delimitation, many wetland ecosystems are temporarily in both categories; this was the best functional definition we found to conceptualize the main message of the MS.

Carbon leaks from flooded land: do we need to re-plumb the inland water active pipe?

Gwenaël Abril1,2 and Alberto V. Borges3

1 Biologie des Organismes et Ecosystèmes Aquatiques (BOREA), UMR 7208, Muséum National d'Histoire Naturelle, CNRS, SU, UCN, UR, IRD, 61 rue Buffon, 75231, Paris cedex 05, France.

2 Programa de Biologia Marinha e Ambientes Costeiros, Universidade Federal
Fluminense, Outeiro São João Batista s/n, 24020015, Niterói, RJ, Brazil.
3 Université de Liège, Unité d'Océanographie Chimique, Institut de Physique (B5a), B-

4000, Belgium

MS for Biogeosciences Discussions, Article type: Ideas and perspectives

**ABSTRACT**

At the global scale, inland waters are a significant source of atmospheric carbon (C), particularly in the tropics. The active pipe concept predicts that C emissions from streams, lakes and rivers are largely fuelled by terrestrial ecosystems. The traditionally recognized C transfer mechanisms from terrestrial to aquatic systems are surface runoff and groundwater drainage. We present here a series of arguments that support the idea that land flooding is an additional significant process that fuels inland waters with C at the global scale. Whether the majority of CO2 emitted by rivers comes from floodable land (approximately 10% of the continents) or from well-drained land is a fundamental question that impacts our capacity to predict how these C fluxes might change in the future. Using classical concepts in ecology, we propose, as a necessary step forward, an update of the active pipe concept that differentiates floodable land from drained land. Contrarily to well-drained land, many wetlands (in particular riparian and littoral wetlands) combine strong hydrological connectivity with inland waters, high productivity assimilating CO2 from the atmosphere, direct transfer of litter and exudation products to water and waterlogged soils, a generally dominant allocation of ecosystem respiration below the water surface and a slow gas exchange rate at the water-air interface. These properties force plants to pump atmospheric C to wetland waters and, when hydrology is favourable, to inland waters as organic C and dissolved CO2. This wetland CO2 pump may contribute disproportionately to CO2 emissions from inland waters, particularly in the tropics where 80% of the global CO2 emissions to the atmosphere occur, In future studies, more care must be taken in the way that vertical and horizontal C fluxes are conceptualized along watersheds and 2D-models that adequately account for the hydrological export of all C species are necessary. In flooded ecosystems, significant effort should be dedicated to quantifying the components of

ecosystem community, and using these metabolic rates in coupled hydrologicalbiogeochemical models. The construction of a global typology of wetlands that includes

productivity, gas fluxes and hydrological connectivity with inland waters also appears

necessary to adequately integrate continental C fluxes at the global scale.

**1. INTRODUCTION**

[revised manuscript text omitted]

| 1 | gwen abril 9/12/18 21:11 |
|---|--------------------------|
|   | gwen abril 9/12/18 21:11 |
|   | gwen abril 9/12/18 21:11 |

| -        | gwen abril 1/11/18 19:49  |
|----------|---------------------------|
| -        | gwen abril 1/11/18 19:49  |
| Υ        | gwen abril 29/12/18 15:30 |

Hamilton et al. 1995) and in flooded forest (Piedade et al. 2010). Total AR in flooded

ecosystems should be divided into three components according to:

**AR=ARa+ARw+ARs\_(Eq. 2)**

where ARa, ARw and ARs are the fraction of AR occurring in air, water and soils, respectively (Fig. 3). In flooded land, a canopy of vegetation generally protects the water-air interface from wind stress and the gas transfer velocity is lower compared to surrounding open waters (Foster-Martinez and Variano 2016; Ho et al. 2018). Consequently, only a limited fraction of ARw and ARs will contribute to the CO2 fluxes measured with static chambers in wetlands. This is a second reason why wetland mass balances are incomplete and may artificially shift wetlands to atmospheric C sources or sinks (Sjögersten et al. 2014).

The allocation of C stocks and metabolism above and below water is fundamentally different in flooded land compared to well-drained land, and this considerably modifies their ecological functionalities (Fig. 2 and 3). Although some wetland plants also use DIC from water for photosynthesis, a large majority of wetland GPP is made by the emerged part of plants that fix atmospheric CO2 during the emersion periods, and/or during the flooding thanks to their emerged or floating canopies (Piedade et al. 1994; Parolin et al. 2001; Engle et al. 2008). A large fraction (excluding wood) of the wetland biomass produced annually is transferred directly to water and sediment as litter fall and fine root production, where it fuels HR, including methanogenesis. Beside some important CH4 oxidation (Segarra et al. 2015), this leads to a Fother (Eq. 1) as CH4 fluxes more significantly in wetlands than in well-drained terrestrial ecosystems (Ciais et al. 2013; Saunois et al. 2016). In addition, because of anaerobic conditions in their soils, water-tolerant plants can develop morphological aeration strategies (Haase and Rätsch 2010)

All these observations suggest the occurrence of a *wetland CO2 pump* that captures atmospheric CO2 and exports organic and inorganic C to rivers and lakes, This biological pump is also consistent with chamber measurements that generally identify CO2 sinks in vegetated flooded areas and CO2 sources in adjacent open waters (Pierobon et al. 2011; Ribaudo et al. 2012; Peixoto et al. 2016). It is worth noting that little is known on how wetland -NEE is affected by hydrology. For instance, a swamp of papyrus (*Cyperus papyrus*) on a sheltered shore of Lake Naivasha, Kenya, was a CO2 sink during immersion but a CO2 source during emersion, when large amounts of plant detritus accumulated in

33

Concerning the metabolic C balance of wetland during flooding, the fraction of OC produced by NPP that is not respired *in situ* or buried in the wetland soil is exported to rivers systems as OC (Fig. 3), according to:

NPP = B+HR+ $E_{POC}$ + $E_{DOC}$  (Eq. 10)

NEP =  $B + E_{POC} + E_{DOC}$  (Eq. 11)

where B is the OC burial in the wetland soil. Thus, the export of POC and DOC from wetlands is expressed as:

 $E_{POC}+E_{DOC} = NEP - B = NPP - HR - B (Eq. 12)$

Downstream, this organic material will undergo intense degradation in inland water (negative NEP), contributing to CO2 outgassing through the OC detrital pathway (Cole and Caraco 2001; Battin et al. 2008).

Plants and microbes respiring in water, sediments, and the root zone (ARw and ARs and HR) release dissolved  $CO_2$  in wetland water. During flooding, ARa is the only component of ER not contributing to  $E_{CO2}$ . The fraction  $\alpha$  of wetland ER occurring in water and sediment (ARw and ARs) and almost all of the microbial HR (Eq. 11), release dissolved  $CO_2$  (and CH4) to waters:

gwen abril 9/12/18 21:18
gwen abril 9/12/18 21:18

 $\alpha$ ER= ARw+ARs+HR

(Eq. <mark>13</mark>)

gwen abril 9/12/18 21:18

part of these dissolved gases are emitted to the atmosphere, and another part is exported by the water flow:

 $\alpha$ ER= FCO2+FCH4+ECO2+ECH4 (Eq. 14)

|                                                                   | $\mathbf{E} = \mathbf{e} (\mathbf{E} \mathbf{E} - \mathbf{e} \mathbf{E} \mathbf{E} - \mathbf{e} (1 - \mathbf{e}) \mathbf{E} \mathbf{E} - \mathbf{e} \mathbf{E} (1 - \mathbf{e} \mathbf{E})$ |                            | Supprimé: 12                                            |
|-------------------------------------------------------------------|---------------------------------------------------------------------------------------------------------------------------------------------------------------------------------------------|----------------------------|---------------------------------------------------------|
| with                                                              | $E_{CO2} = \alpha \beta E R$ and $F_{CO2} = \alpha (1 - \beta) E R$ and $(0 < \beta < 1) (Eq. 15)$                                                                                          |                            | gwen abril 9/12/18 21:18                                |
| $\alpha\beta$ is thus the                                         | e fraction of ecosystem respiration that is exported laterally fr                                                                                                                           | om the                     | Supprimé: 13                                            |
| ,                                                                 |                                                                                                                                                                                             |                            |                                                         |
| wetland in w                                                      | ater masses. For simplification, we do not include E CH4 in Eq. 1                                                                                                         | 3 because           | gwen abril 1/11/18 19:56                                |
| this term is a                                                    | ssumed to be modest (few %) compared to E_{CO2}. Indeed, the $eta$                                                                                                                          | term might                 | Supprimé: this last equation                            |
| be much sma                                                       | ller for $CH_4$ than for $CO_2$ due to preferential $CH_4$ ebullition and                                                                                                                   | transport                  |                                                         |
| through plan                                                      | ts in wetlands (Chanton and Whiting 1995). For $CO_2$ , the fraction                                                                                                                        | on $\beta$ depends         |                                                         |
| on hydrologi                                                      | cal and geomorphological parameters such as water depth, vel                                                                                                                                | ocity and gas              |                                                         |
| exchange in t                                                     | he wetland. Using a simple model of lateral dissolved gas tran                                                                                                                              | sport (Abril               |                                                         |
| et al. 2014), t                                                   | ypical values of 1 cm s $^{-1}$ for the gas transfer velocity (Foster-M                                                                                                                     | lartinez and               |                                                         |
| Variano 2016                                                      | 6; Ho et al. 2018) and 5000 ppmv for water pCO 2 , we calculated                                                                                                                 | d a $β$ value of           |                                                         |
| 0.93 for a wa                                                     | ter column of 1 m-depth flowing at a velocity of 10 cm s $^{-1}$ in a                                                                                                                       | 100 m-long                 |                                                         |
| wetland (ass                                                      | umed conditions for riparian wetlands during maximum flood                                                                                                                                  | . When the                 |                                                         |
| water depth i                                                     | is <mark>set</mark> at 0.1 m instead of 1 m or the water velocity is establishe                                                                                                             | ed at 1 cm s -1 |                                                         |
| instead of 10                                                     | cm s -1 , $\beta$ decreases to 0.53. Consequently, a large majority of t                                                                                                         | he CO 2         | Supprimé: established                                   |
| produced by                                                       | wetland below-water respiration is outgassed to the atmosph                                                                                                                                 | ere outside of             |                                                         |
| the wetland.                                                      | Finally, accounting for all terms in Eq. 6 in wetlands leads to to                                                                                                                          | otal export                |                                                         |
| expressed as                                                      | :                                                                                                                                                                                           |                            |                                                         |
| $E = (E_{DOC} + E_{PC})$                                          | $(E_{CO2}+E_{CH4})=(NPP-HR-B)+(\beta \alpha ER -FCO_2-FCH_4)$                                                                                                                               | (Eq. 16 )           | awan ahril 0/12/19 21-19                                |
| $\mathbf{E} = (\mathbf{E}_{\text{DOC}} + \mathbf{E}_{\text{PC}})$ | $(E_{CO2}+E_{CH4})=(NPP-HR-B)+(\beta(ARw+ARs+HR)-FCO_2-FCH_4)$                                                                                                                              | (Eq. <mark>17</mark> )     | Gupprimé:         14           gwen abril 9/12/18 21:18 |
| $E = NPP-B+\beta$                                                 | ARw+ $\beta$ ARs+( $\beta$ – 1)HR-FCO 2 -FCH 4                                                                                                                        | (Eq. <mark>18</mark> )     | Supprimé: 14                                            |

[revised manuscript text omitted]